# Effects of supplemental feeding on the fecal bacterial communities of Rocky Mountain elk in the Greater Yellowstone Ecosystem

**Claire E. Couch**[1]*, **Benjamin L. Wise**[2], **Brandon M. Scurlock**[2], **Jared D. Rogerson**[3], **Rebecca K. Fuda**[4], **Eric K. Cole**[5], **Kimberly E. Szcodronski**[6], **Adam J. Sepulveda**[6], **Patrick R. Hutchins**[6], **Paul C. Cross**[6]

**1** Department of Fisheries & Wildlife, Oregon State University, Corvallis, Oregon, United States of America, **2** Wyoming Game & Fish Department, Jackson, Wyoming, United States of America, **3** Wyoming Game & Fish Department, Pinedale, Wyoming, United States of America, **4** Oregon Department of Fish & Wildlife, Prineville, Oregon, United States of America, **5** U.S. Fish & Wildlife Service, National Elk Refuge, Jackson, Wyoming, United States of America, **6** U.S. Geological Survey, Northern Rocky Mountain Science Center, Bozeman, Montana, United States of America

* claire.couch@oregonstate.edu

**Data Availability Statement:** Sequencing data are available in the NCBI Sequence Read Archive and

## Abstract

Supplemental feeding of wildlife is a common practice often undertaken for recreational or management purposes, but it may have unintended consequences for animal health. Understanding cryptic effects of diet supplementation on the gut microbiomes of wild mammals is important to inform conservation and management strategies. Multiple laboratory studies have demonstrated the importance of the gut microbiome for extracting and synthesizing nutrients, modulating host immunity, and many other vital host functions, but these relationships can be disrupted by dietary perturbation. The well-described interplay between diet, the microbiome, and host health in laboratory and human systems highlights the need to understand the consequences of supplemental feeding on the microbiomes of free-ranging animal populations. This study describes changes to the gut microbiomes of wild elk under different supplemental feeding regimes. We demonstrated significant cross-sectional variation between elk at different feeding locations and identified several relatively low-abundance bacterial genera that differed between fed versus unfed groups. In addition, we followed four of these populations through mid-season changes in supplemental feeding regimes and demonstrated a significant shift in microbiome composition in a single population that changed from natural forage to supplementation with alfalfa pellets. Some of the taxonomic shifts in this population mirrored changes associated with ruminal acidosis in domestic livestock. We discerned no significant changes in the population that shifted from natural forage to hay supplementation, or in the populations that changed from one type of hay to another. Our results suggest that supplementation with alfalfa pellets alters the native gut microbiome of elk, with potential implications for population health.

publicly accessible under BioProject ID
PRJNA629905.

**Funding:** Funding for this project was provided by
the NSF Graduate Research Internship Program
(https://www.nsf.gov/funding/pgm_summ.jsp?
pims_id=505127, awarded to CEC) and the USGS
Ecosystems and Environmental Health Mission
Areas (https://www.usgs.gov/mission-areas/
environmental-health, awarded to PCC). The
funders had no role in study design, data collection
and analysis, decision to publish, or preparation of
the manuscript.

**Competing interests:** The authors have declared
that no competing interests exist.

## Introduction

Supplemental feeding of wildlife is a widespread but controversial practice that occurs at
human-wildlife interfaces across the globe [1]. Feeding may be undertaken for recreational
purposes such as wildlife viewing [2] or hunting [3], or for management purposes such as
increasing population density [4] or diverting wildlife movement and feeding patterns to
reduce conflict [5]. However, feeding can have unintended consequences such as increased
disease transmission [5] or altered species interactions [6]. Understanding how supplemental
feeding impacts cryptic aspects of host health is key to optimizing conservation and manage-
ment decisions as the human-wildlife interface continues to expand and change.

In the past two decades, multiple studies in humans and domestic animals have shown that
diet is a key driver of variation in the gut microbiome [7–10], and aspects of this variation
have in turn been shown to associate with host health and disease [11,12]. The gut microbiome
plays crucial roles in multiple host functions including nutrient extraction [13], immunity
[14], and hormone regulation [15]. An emerging body of research is beginning to suggest
intriguing patterns of microbiome variation in wild mammalian populations that may be
relevant to conservation [16,17]. There is potential for microbiomes to serve as a tool for con-
servation efforts such as surveying population health and immunity [18], understanding con-
nectivity between individuals [19] and populations [20], and improving survival prospects of
reintroduced or translocated individuals [21–23]. However, there is little research directly con-
necting current management actions, microbiome dynamics, and consequences for host
health. The impacts of supplemental feeding programs on the microbiomes of wildlife popula-
tions have been directly addressed in only a few studies [24,25], and their findings underscore
the importance of clarifying the links between anthropogenic diet inputs, gut microbiome
shifts, and downstream impacts on wildlife health.

Covariation between diet and gut microbiome in wildlife depend largely on host phylogeny
and environment [26,27]. Temporal variation in gut microbiome communities has been
shown to correlate with seasonal fluctuations in diet composition within wild mammalian
populations [24,28,29]. However, because diet, social structure, and environmental conditions
such as temperature and precipitation often covary alongside seasonal dietary changes, it can
be challenging to disentangle their relative effects on microbiome communities. A number of
studies have identified gut microbiome discrepancies between captive versus wild populations
of conspecific mammals [30–32], presumably related to differences in diet. Again though, it is
difficult to determine whether diet or one of the other manifold environmental or social differ-
ences between captive versus wild populations drives these differences. Findings from the few
studies that have addressed the impacts of supplemental feeding in wild populations support
the hypothesis that feeding can significantly alter gut community structure in wild hosts
[24,25,33]. Because supplemental feeding is a widely used management strategy that is often
intended to increase population numbers or reduce human-wildlife conflict, understanding
the consequences of supplemental feeding on gut microbiome communities in wildlife, and
the consequences for host health, would be of great value to wildlife managers.

Each winter, elk in the Greater Yellowstone Ecosystem are provided with supplemental
feed at more than 20 locations (feedgrounds) throughout western Wyoming. Most state-oper-
ated feedgrounds provide loose grass, alfalfa, or mixed alfalfa/grass hay beginning in December
or January, depending on snowfall conditions, and continues until elk disperse to seek spring-
time forage in March or April [34]. On the U.S. Fish and Wildlife Service's National Elk Refuge
(NER) near Jackson, WY, USA, elk are provided with compressed alfalfa pellets which provide
more concentrated nutritional value than loose hay. Supplemental feeding of elk is highly con-
troversial. Although feeding can mitigate human-wildlife conflict by reducing comingling with

livestock and can support large populations in lieu of native habitat, there is concern that feed-grounds act as hotspots for disease transmission [35,36]. Research on the impacts of feed-grounds on disease dynamics in this system is ongoing [37], but other cryptic impacts of feeding, including potential impacts on gut microbiota, have not been explored.

In this study, we assess the impacts of supplemental winter feeding on gut microbiome dynamics among Rocky Mountain elk (*Cervus canadensis nelsoni*) attending feedgrounds in western Wyoming by describing commensal gut microbiome variation related to supplemental feeding regimes and exploring potential implications for elk population health and disease. We compared cross-sectional samples from active feedgrounds and unfed control groups and assessed longitudinal changes in four of these populations that experienced mid-season changes to feeding regime. We hypothesized that microbiome composition would differ based on feed type in the cross-sectional comparison, and that compositional shifts would correlate with feed regime changes in the longitudinal study. Additionally, we explored possible correlations between diet-driven microbiome changes and elk population health and disease. As part of this exploratory work, we developed an elk-specific assay to assess prevalence and abundance of *Fusobacterium necrophorum*, a ubiquitous resident of ruminant gastrointestinal (GI) tract microbiomes that has been linked with hoof rot and necrotizing stomatitis in ruminants [38,39] and is a pathogen of concern for elk on Wyoming feedgrounds [40]. Overall, we sought to describe diet-driven alterations to the gut microbiome and identify priorities for future research linking the gut microbiome with elk population health.

## Materials & methods

### Sample collection & storage

Fresh fecal pellets were collected from elk at twelve feedgrounds and two native winter range sites (Fig 1, Table 1). GPS collar data demonstrates that the vast majority of elk remain at a single feeding location for the duration of winter, rarely dispersing more than 5 km [41], For elk on feedgrounds, sampling was conducted noninvasively from the ground or by habituating elk to feeding in corrals for capture and then directly collecting feces from the rectum. Sub-freezing temperatures typical of western Wyoming during the feeding season enabled us to assess freshness of noninvasively collected feces. We collected samples that were still warm and moist from snow-covered ground and assumed that, under sub-freezing conditions, these samples were likely less than one hour old. Elk are estimated to defecate approximately once every 2–2.5 hours while grazing [42], therefore we assumed that samples at each time point came from different individuals. Fecal samples were collected using sterile gloves and placed in individual whirl-pack sample bags or 50 ml conical tubes. For elk on native winter range, samples were opportunistically collected directly from the rectum when animals were captured via net guns for collaring. On the NER, samples from the first time points (including the cross-sectional time point) were collected noninvasively prior to the initiation of feeding operations, and the following three time points were collected at two, four, and six weeks after feeding commenced. Cross-sectional samples were collected between January 20–27, 2019, and longitudinal samples were collected opportunistically from November 2018-April 2019. Between 8–23 samples were collected per location per time point and stored at -20 ˚C until processing (Table 1). Hay samples were collected concurrently with cross-sectional fecal samples from feedgrounds, and alfalfa pellet samples were obtained from the NER after feeding commenced. Hay and alfalfa pellet samples were outsourced for nutrient content analysis (A&L Western Laboratories, Modesto, CA). Samples from state-run feedgrounds and native range elk were collected under the supervision of Wyoming Game and Fish Department during routine

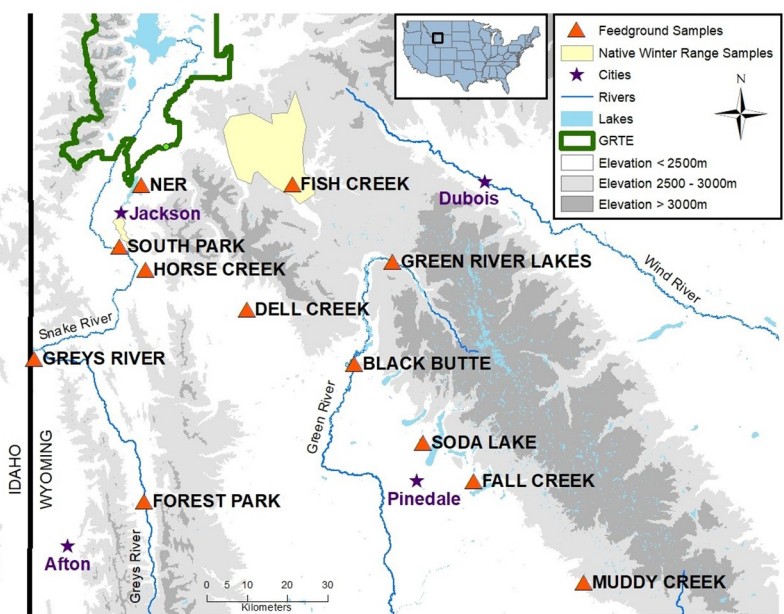

**Fig 1. Geographic locations of elk microbiome sample collection sites.** This study included elk from twelve feedgrounds (orange triangles), in addition to unfed elk on native winter range (filled yellow polygons). Longitudinal samples were collected from South Park, Horse Creek, Fish Creek, and the National Elk Refuge (map courtesy of the U.S. Geological Survey).

monitoring and captures, and samples from the NER were collected by USFW personnel during routine monitoring, therefore no project-specific permits were required.

## Sample processing & sequencing

DNA extraction, PCR amplification, and 16S sequencing were performed by the Center for Genome Research & Biocomputing at Oregon State University. For each sample, a single fecal pellet was homogenized, and then a 200 mg aliquot was used for DNA extraction according to the Earth Microbiome Protocol [43]. PCR and sequencing of the 16S V4 region were performed according to the Earth Microbiome protocol using amplification primers 515F and 806R [44,45]. Samples were split equally between two MiSeq runs that included a total of 315 elk fecal samples, including the 282 samples used in this study. Details of the *F. necrophorum* qPCR assay development and validation are provided in S1 Appendix and S2 Table. In addition to nonspecific 16S sequencing, we ran a targeted qPCR assay to detect two subspecies of *F. necrophorum* (ssp. *necrophorum* and ssp. *funduliforme*) [46] while normalizing based on host DNA content [47] (see S1 Appendix for methods).

## Statistical analysis

DADA2 (version 1.12.1) was used to identify amplicon sequence variants (ASVs), trim adapter sequences, and remove chimeras [48]. Raw sequence data were processed through the DADA2 pipeline using the following trimming parameters: truncLen = c(240, 200), maxN = 0, maxEE = c(2,2), truncQ = 2, rm.phix = TRUE. Default parameters were used for estimating error parameters using learnErrors(), and chimeras were removed using removeBimeraDenova (method = "consensus"). A total of 22,620,453 reads were obtained from 315 samples following initial preprocessing steps. Prior to statistical analyses, samples with less than 20,000

**Table 1. Distribution of elk fecal samples across time, space, and sampling methodologies.**

| Location | Date | Collection method | Number of samples | Feed type |
|---|---|---|---|---|
| Fish Creek | 12/20/2018 | Corral | 8 | Alfalfa/grass hay mix |
| | 1/7/2019 | Corral | 23 | Alfalfa/grass hay mix |
| | 1/20/2019 | Noninvasive | 10 | Alfalfa/grass mix |
| Horse Creek | 1/14/2019 | Noninvasive | 10 | Grass hay |
| | 1/22/2019 | Noninvasive | 19 | Grass hay |
| | 4/4/2019 | Noninvasive | 9 | Alfalfa hay |
| South Park | 1/14/2019 | Noninvasive | 10 | Grass hay |
| | 1/22/2019 | Noninvasive | 10 | Grass hay |
| | 3/11/2019 | Noninvasive | 10 | Alfalfa hay |
| | 4/4/2019 | Noninvasive | 10 | Alfalfa hay |
| National Elk Refuge (NER) | 1/21/2019 | Noninvasive | 10 | Natural |
| | 2/4/2019 | Noninvasive | 10 | Natural |
| | 2/24/2019 | Noninvasive | 10 | Alfalfa pellets |
| | 3/8/2019 | Noninvasive | 10 | Alfalfa pellets |
| | 3/24/2019 | Noninvasive | 10 | Alfalfa pellets |
| Black Butte | 1/22/2019 | Noninvasive | 10 | Grass hay |
| Green River Lakes | 1/23/2019 | Noninvasive | 10 | Grass hay |
| Soda Lake | 1/25/2019 | Noninvasive | 10 | Grass hay |
| Grey's River | 1/19/2019 | Noninvasive | 9 | Alfalfa/grass mix |
| Forest Park | 1/19/2019 | Noninvasive | 10 | Alfalfa/grass mix |
| Alpine | 1/19/2019 | Noninvasive | 9 | Alfalfa/grass mix |
| Muddy Creek | 1/22/2019 | Noninvasive | 10 | Alfalfa hay |
| Dell Creek | 1/24/2019 | Noninvasive | 10 | Alfalfa hay |
| Fall Creek Feedground | 1/24/2019 | Noninvasive | 10 | Alfalfa hay |
| South Jackson Native Winter Range (near South Park) | 1/27/2019 | Net | 9 | Natural |
| Gros Ventre Native Winter Range (near Fish Creek) | 11/5/2018 | Net | 16 | Natural |

Samples from elk on native winter range were obtained directly from the animals following net-gun capture (Net). Samples from elk on feedgrounds were collected either from the ground (Noninvasive), or by habituating elk to feeding in an enclosure for capture and direct sampling (Corral).

reads were removed, and the remaining 282 samples were rarified to the minimum sequencing depth of 29,710 reads per sample. All statistical analyses were performed in R version 3.6.3 unless otherwise specified [49].

Microbiome richness was calculated as number of unique ASVs in each sample. Richness and relative taxonomic abundance from phylum-genus ranks were calculated and visualized in the phyloseq package (version 1.30.0) [50]. Inverse Simpson and Shannon diversity indices, both of which incorporate taxonomic evenness in addition to richness, were also calculated in phyloseq [50] to assess whether alpha diversity results were especially sensitive to changes in rare taxa (Shannon) or common taxa (Inverse Simpson). Cross-sectional variation in microbiome richness across feed regimes was assessed using generalized linear mixed models (GLMMs) with feed as a categorical fixed effect and location as a random effect. This model was compared to a null model containing only location as a random effect using a chi-squared test. Both models were generated using the lmer function in the lme4 package based on a Poisson distribution. Results from the richness model were verified using Kruskal-Wallis tests and pairwise Wilcoxon rank-sum tests. For the Inverse Simpson and Shannon diversity metrics, GLMMs with a random effect for location resulted in singularities due to insufficient variance among locations, therefore we relied on Kruskal-Wallis tests and pairwise Wilcoxon rank-sum

tests to assess diet-associated variation in these indices. For all pairwise Wilcoxon rank-sum tests, we applied false discovery rate (FDR) correction to resulting p-values.

For microbiome compositional analysis, ASVs were merged by genus for ease of interpretation and to reduce computational intensity. In order to assess inter-group variation among feed types and sampling locations, the nested.npermanova function in the BiodiversityR package (version 2.11–3) [51] was used to perform nested PERMANOVA tests with sample location nested within feed type. To visualize compositional differences between feed types, principal coordinate analysis (PCoA) was run on Bray-Curtis distances between all cross-sectional samples using the ordinate function in phyloseq, and the first three axes were plotted using plot_ordination. To identify taxa that differed significantly between fed versus unfed elk, we used the linear discriminate analysis effect size (LEfSe) approach [52]. Briefly, this method performs non-parametric tests between classes (i.e. feed status) that are consistent among subclasses (i.e. location) to identify significantly different taxa, and then uses linear discriminate analysis to estimate the effect size of each differentially abundant taxon. Taxa that showed significant differences between classes, and were consistent among subclasses, were reported if the effect size was greater than log 2-fold between the two classes. To assess longitudinal changes in microbiome communities associated with changes in diet regime within the four longitudinally sampled populations, we performed PCoA on sample-wise Bray-Curtis distances for each population for visualization and performed nested PERMANOVA tests with collection date nested within feed type. In the population where diet change significantly associated with microbiome shifts, we used LEfSe to identify the significantly different taxa from the phylum through genus levels as described above but using diet as class and collection date as subclass.

## Results

### Cross-sectional variation in alpha diversity and composition

Among the cross-sectional samples, microbiome richness ranged from 434–1299 unique amplicon sequence variants (ASVs) per sample. The GLMM that included feed type as a fixed effect was significantly better than the location-only null model (chi-square p = 0.0028), indicating that feed regime is a significant driver of variation in richness between locations. A pairwise Wilcoxon rank-sum test supported these results, indicating that richness was significantly lower among unfed compared with supplementally fed elk (p < 0.005 for each pairwise comparison), but that among fed elk, different feed types did not significantly impact richness (Fig 2). Pairwise Wilcoxon rank-sum tests also demonstrated significantly lower alpha diversity index values in unfed versus fed elk (Inverse Simpson FDR-corrected p <0.05 and Shannon FDR-corrected p <0.001 for each pairwise comparison). The top twenty most abundant genera across all populations belonged to seven families comprised in three unique orders representing phyla Firmicutes, Bacteroidetes, and Verrucumicrobia (Fig 2). *F. necrophorum funduliforme* was identified in only a single fecal sample, and *F. necrophorum necrophorum* was not detected in any samples, suggesting that these species are rarely or never shed in elk feces.

### Diet-related microbiome differences among populations

Significant compositional differences between location were observed (p = 0.0001) based on nested PERMANOVA tests, but differences between feed type were marginal (p = 0.07). Based on visualization of PCoA axes 1–2, which collectively accounted for 41.5% of the variance among samples, differences among feed types were not visually apparent (Fig 3). This pattern aligns with Fig 2, which shows no obvious compositional differences related to feed type in the

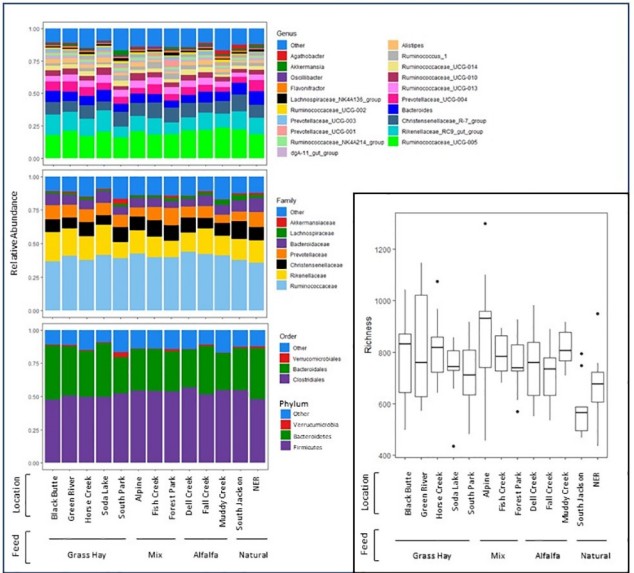

**Fig 2. Relative abundance for the top 20 most abundant genera are shown for each population in the cross-sectional study.** Fill color indicates genus (top left), family (middle left) or order/phylum (bottom left). Amplicon sequence variant-level richness and alpha diversity for each population is shown in the inset (bottom right) where the midline indicates the median values, hinges indicate the first and third quartiles, whiskers extend up to 1.5 the interquartile range, and outliers beyond this range are represented as individual points. Populations are grouped by diet at the time of sample collection. Note that cross-sectional samples from the National Elk Refuge (NER) were collected prior to commencement of feeding at that location.

most abundant genera. However, separation along PCoA axis 3, which accounted for 8.2% of the variance among samples, suggested that unfed elk differed from other feed regimes along that axis. In support of this finding, LEfSe revealed that several low-abundance taxa significantly differed among fed vs unfed elk after accounting for location (Fig 4). Genus *Ruminococcaceae UCG-009* was enriched in fed elk, whereas genera *Erysipelatoclostridium* and *Flexilinea* (and parent clades through the phylum level) were enriched in unfed elk (Fig 4).

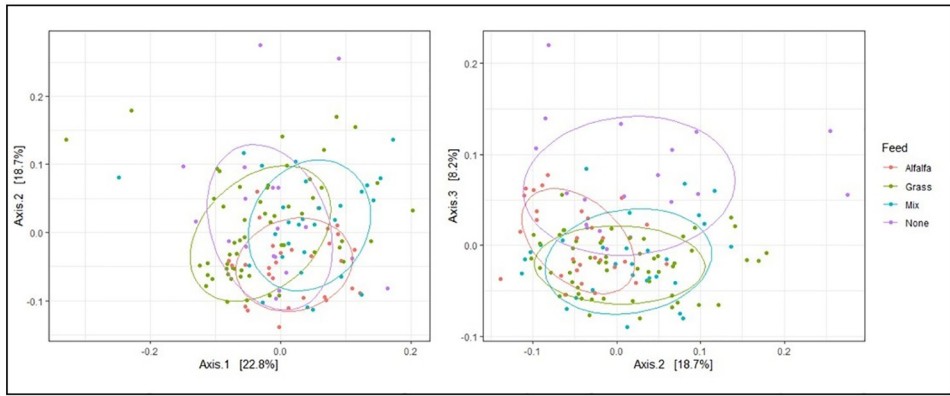

**Fig 3. Principal coordinate analysis of cross-sectional elk gut microbiomes sampled from 13 locations, including 11 feedgrounds stratified among three different feed types and two unfed control groups.** The left panel shows the first two principal coordinate axes, and the right panel shows the second and third axes. Collectively, the first three axes explained 49.7% of the variance among samples. Ellipses are drawn around 70% of the data points in each feed group.

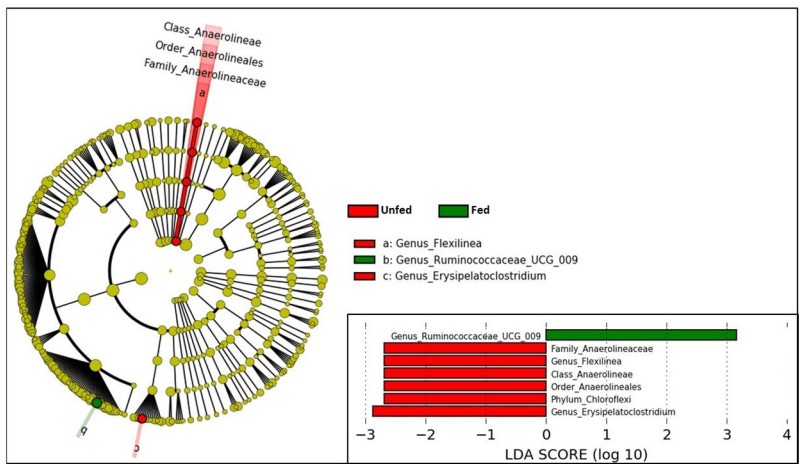

**Fig 4. Differentially abundant microbial taxa between fed and unfed elk.** Linear discriminate analysis was used to identify taxa (phylum-genus levels) that exhibited log-two-fold abundance changes between the two groups.

## Longitudinal microbiome shifts related to feed change

We longitudinally sampled elk at 4 different locations before and after feed regime transitions to determine whether the change in feed type resulted in a change to microbial communities. In the longitudinal series, CCA and nested PERMANOVA tests revealed significant differences between pre- and post-feed samples only within the NER population, which transitioned from no supplemental feed ("natural diet") to pelleted alfalfa (Fig 5). No significant taxonomic changes were identified in the population that transitioned from natural diet to alfalfa/grass mix (Fish Creek) or in either of the populations that transitioned from grass hay to alfalfa hay (South Park & Horse Creek). Nutrient analysis demonstrated that that pelleted alfalfa had lower fiber content and somewhat lower NFE (soluble carbohydrates), but higher protein and

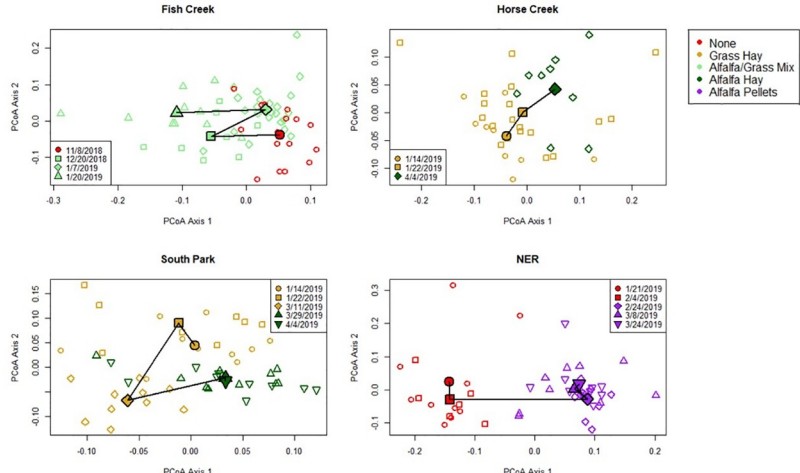

**Fig 5. Principal coordinate analysis plots of gut microbiome samples from populations that underwent mid-season shifts in feeding regime over the course of the study.** Principal coordinate analysis was performed on Bray-Curtis distances separately for each population, therefore axes do not represent the same dimensions for each plot. Individual samples are represented by unfilled symbols, and population centroids for each time point are represented by filled symbols. Color represents feed regime, and shape indicates sample date. Centroids for each sampling date are connected in temporal order with solid lines to highlight compositional shifts over time.

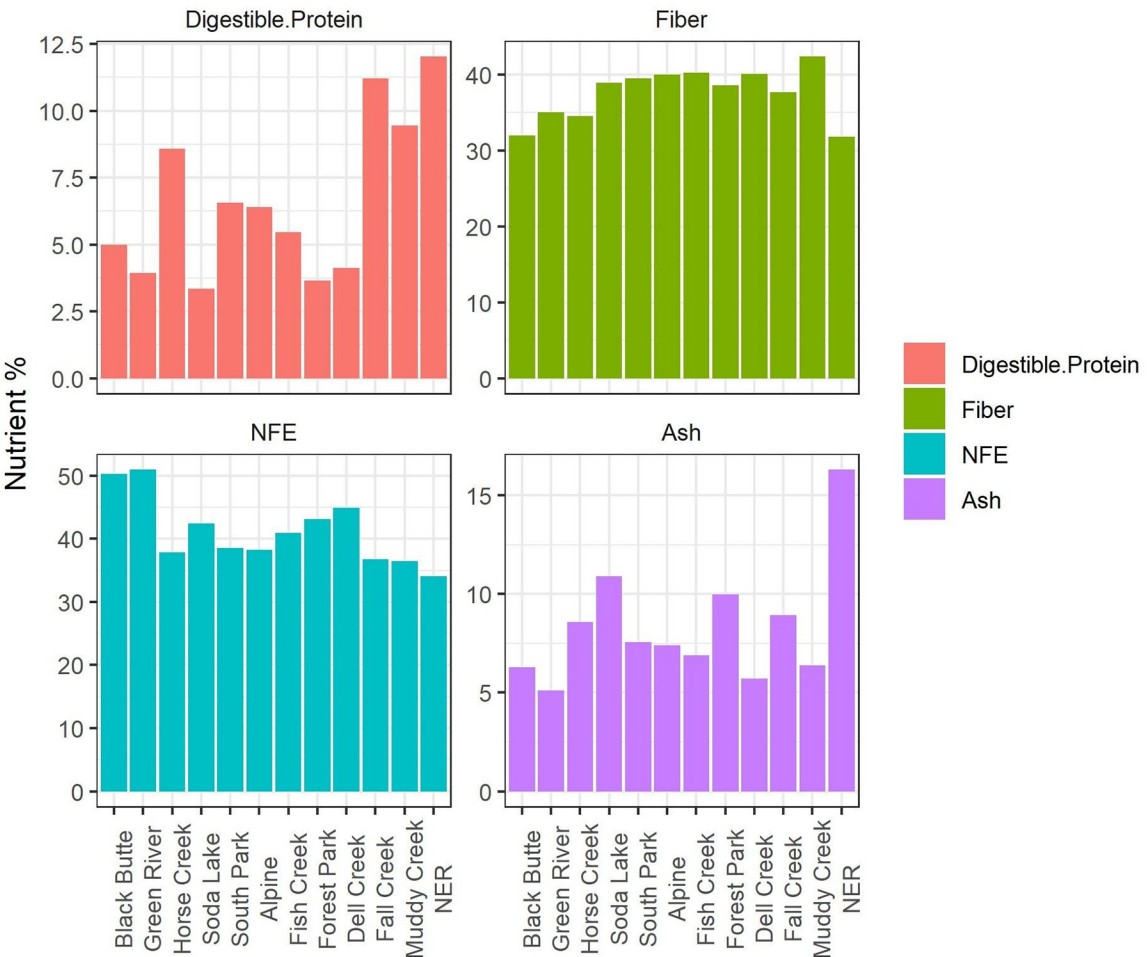

**Fig 6. Comparison of macronutrient levels in the hay and pellets fed to elk at the Wyoming feedgrounds in this study.** We measured percentages of digestible protein, fiber, nitrogen-free extract (soluble, non-fiber carbohydrates), and ash (total mineral content). Feed samples from Horse Creek and South Park were collected before the transition from grass hay to alfalfa hay, and feed samples from the National Elk Refuge were collected following the commencement of feeding operations in the longitudinal study.

ash (mineral) content than any of the supplemental hay for which nutrient analysis was conducted (Fig 6, see S1 Table for full nutrient analysis results). The compositional shift in the NER population was characterized by phylum-level increases in Firmicutes, Spirochaetes, and Tenericutes, and decreases in Bacteroidetes, Chloroflexi, Plantomycetes, Proteobacteria, and Verrucromicrobia (Fig 7). Additional shifts at lower taxonomic levels are shown in S1 Fig.

## Discussion

We assessed cross-sectional and longitudinal variation in microbiome diversity and composition among wild elk under supplemental feeding regimes compared with those under natural foraging conditions. Our results suggest that feeding supplemental loose hay (grass, alfalfa, or mix) associates with changes to only a few low-abundance taxa, and that location is more predictive of gut microbiome than feeding regime for hay-based supplemented diets. In contrast, feeding concentrated alfalfa pellets appears to generate significant shifts in gut microbiome composition compared to natural foraging conditions on the National Elk Refuge, such that

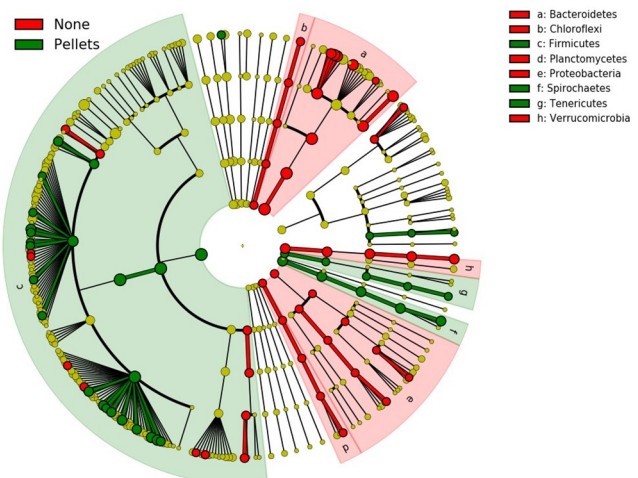

**Fig 7. Linear discriminate analysis effect size (LEfSe) results showing bacterial lineages that differed significantly in a population before and after supplemental feeding with concentrated alfalfa pellets.** Each node in the tree represents a taxon from phylum level through genus level (tips). Nodes are colored red if they were enriched under the natural diet regime, and green if they were enriched under the pelleted feed regime. Yellow nodes did not differ significantly across regimes. Branches are highlighted red if they belong to phyla that were enriched under the natural diet regime, and green if they belong to phyla that were enriched in pellet-fed elk.

these microbiome shifts associated with concentrated feed could have implications for elk population health.

Microbiome communities across locations in the cross-sectional study demonstrated compositional patterns consistent with those reported in studies of other wild ungulates. Across populations, the top twenty bacterial genera were predominantly from phylum Firmicutes, followed by Bacteroidetes, with a small proportion from phylum Veruccumicrobia. This result aligns with microbiome composition from fecal samples previously described in elk [53]. Richness and alpha diversity were lower in unfed elk relative to fed elk, but beta diversity was not significantly associated with diet after controlling for location. The number of population replicates in the cross-sectional study was limited, and future studies should include additional replicates from each feed group, particularly with more samples from unfed control animals, in order to robustly assess the impacts of hay supplementation on microbiome alpha diversity and composition. Although overall beta diversity estimates did not depend on diet, a few genera differed between fed and unfed populations. Genus *Ruminococcus UCG-009* was enriched in fed elk, whereas *Flexilinea* and *Erysipelatoclostridium* were enriched in unfed elk. *Ruminococcus UCG-009* has also been shown to be enriched in captive versus wild Pere David's deer [31], and *Flexilinea* and *Erysipelatoclostridium* vary temporally in other wild herbivores [54,55], presumably due to fluctuating forage availability. Overall, these findings suggest that elk gut microbiome composition is relatively robust to dietary changes associated with hay supplementation, but changes to a few key taxa are consistent with patterns identified in studies of other wild herbivore species.

Significant longitudinal shifts in microbiome composition occurred in the NER population after transitioning from a natural diet to supplementation with alfalfa pellets, possibly due to reduced fiber or increased protein or mineral content relative to other supplemental feed types (Fig 6). LEfSE analysis demonstrated significant increases in 38 taxa and decreases in 49 taxa following the transition to supplemental pellets. At the phylum level, taxonomic shifts included a reduction of Bacteroidetes, Chloroflexi, Plantomycetes, Verrucumicrobia, and

Proteobacteria, and an increase in Firmicutes, Spirochaetes, and Tenericutes following the transition from natural diet to supplementation with concentrated alfalfa pellets. Interestingly, a subset of these taxa is associated with host immunity in laboratory systems. For example, members of phylum Bacteroidetes contribute to the development of gut-associated lymphoid tissues [56], activating the T-cell-dependent immune response [57], and other host immune functions [58]. Recent research also suggests that members of Verrucumicrobia have the potential to induce regulatory immunity in horses [59]. Laboratory studies demonstrate that increasing the Firmicutes:Bacteroidetes ratio reduces short-chain fatty acid production, and it is speculated that these shifts could reduce microglial activity and promote prion diseases; however, this association has not been confirmed [60]. It is therefore possible that some of the diet-driven changes in bacterial relative abundance observed in elk on the NER may associate with changes in immune function, but further research is necessary to directly relate immunity with microbiome community structure in elk.

In addition to potential associations with immunity, the gut microbiome dynamics of elk on the NER may offer other insights into the physiological impacts of supplemental feeding. Elk are intermediate (mixed) ruminants with relatively flexible feeding strategies [61] and significant reliance on microbes in the reticulorumen and hindgut for nutrient extraction. In domestic mixed ruminants, the gastrointestinal microbiome adapts rapidly to diet change [62], and some microbial changes induced by dramatic feed alteration have been linked to rumen acidosis [63] The reduction we observed in Proteobacteria and Verrucomicrobia in elk fed alfalfa pellets reflect some of the changes associated with rumen acidosis in the fecal microbiomes of domestic cattle [64], and the increase in Firmicutes mirrors the change in abundance of this phylum in the rumen of cattle with this condition [64]. Due to physiological differences between elk and cattle, we cannot assume analogous rumen acidosis microbiome phenotypes, but this finding warrants further exploration. Previous work has shown that rumen acidosis is a leading cause of death among captive elk [65], and gastritis of unknown etiology was observed in necropsies of approximately 20% of apparently ill, pellet-fed elk during winter feeding operations at the NER from 2009–2013 (L. Jones, prs. comm.). Understanding the impact of supplemental feeding on this syndrome is crucial to informing management practices. More research is needed to characterize the fecal microbiome shifts associated with rumen acidosis in elk, a question which could be addressed by collecting and comparing rumen and microbiome samples in intensively studied wild or captive elk. Future studies should also account for host demographics, including age and sex, which were not included in this study. This information could then be used to assess the impacts of feed on rumen acidosis via noninvasive fecal microbiome sampling.

The elk microbiome is known to vary significantly along the gastrointestinal tract, thus the relative robustness of the fecal microbiome to hay supplementation does not necessarily reflect robustness along the entire GI tract [53]. In domestic ruminants, the foregut microbiome has higher richness and may be more responsive to feed changes than the fecal microbiome [66] (Lourenco et al. 2020). Therefore, while the fecal microbiomes of ruminants are easily sampled noninvasively, they represent only a subset of the complex and variable gastrointestinal tract microbiome and must be interpreted with care. Notably, some commensal GI bacteria are ubiquitous among their hosts but are rarely shed in the feces, and therefore any potential effects of supplemental feeding on these bacteria remain cryptic. Based on a recent study in domestic sheep, this is likely the case with *F. necrophorum* [67], which would account for the non-detection of this widespread ruminant commensal in elk feces. Future studies should assess changes to the microbiome of the rumen and other sites along the GI tract that occur as a result of hay supplementation, including changes in *F. necrophorum* abundance and distribution.

Our work suggests that supplementation with hay (grass, alfalfa, or mix) has a much smaller impact on fecal microbiome composition than concentrated alfalfa pellets. Shifts in microbiome composition observed in an elk population that transitioned from natural feed to supplemental concentrate may be related to immune functioning or to subacute rumen acidosis in elk and therefore warrant further investigation. More broadly, this study underscores the potential of gut microbiome studies as a tool for noninvasive monitoring of population health in wildlife conservation efforts.

## Supporting information

**S1 Fig. LEfSe results showing bacterial taxa that differed significantly in elk gut microbiomes before and after supplementary feeding with concentrated alfalfa pellets.**
(TIF)

**S1 Table. Full nutrient analysis results.**
(DOCX)

**S2 Table. Oligonucleotides used in multiplex qPCR assays for *F. necrophorum*.**
(DOCX)

**S1 Appendix. Design and optimization of *Fusobacterium necrophorum* qPCR assay.**
(DOCX)

## Acknowledgments

We thank Anna Jolles, Thomas Sharpton, Hank Edwards, and Lee Jones for their guidance with manuscript preparation. Any use of trade, firm, or product names is for descriptive purposes only and does not imply endorsement by the U.S. Government.

## Author Contributions

**Conceptualization:** Claire E. Couch, Paul C. Cross.

**Formal analysis:** Claire E. Couch.

**Funding acquisition:** Claire E. Couch, Paul C. Cross.

**Investigation:** Claire E. Couch, Benjamin L. Wise, Brandon M. Scurlock, Jared D. Rogerson, Rebecca K. Fuda, Eric K. Cole, Adam J. Sepulveda, Patrick R. Hutchins.

**Methodology:** Adam J. Sepulveda, Patrick R. Hutchins.

**Supervision:** Paul C. Cross.

**Visualization:** Claire E. Couch, Kimberly E. Szcodronski.

**Writing – original draft:** Claire E. Couch.

**Writing – review & editing:** Benjamin L. Wise, Brandon M. Scurlock, Jared D. Rogerson, Rebecca K. Fuda, Eric K. Cole, Kimberly E. Szcodronski, Adam J. Sepulveda, Patrick R. Hutchins, Paul C. Cross.

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
