## [Decision Letter · Decision Letter 0]

30 Jun 2020

PONE-D-20-13745

Effects of supplemental feeding on the gut microbiomes of Rocky Mountain elk in the Greater Yellowstone Ecosystem

PLOS ONE

Dear Dr. Couch,

Thank you for submitting your manuscript to PLOS ONE. After careful consideration, we feel that it has merit but does not fully meet PLOS ONE’s publication criteria as it currently stands. Therefore, we invite you to submit a revised version of the manuscript that addresses the points raised during the review process.

We look forward to receiving your revised manuscript.

Kind regards,

Juan J Loor

Academic Editor

PLOS ONE

Journal Requirements:

4. We note that Figure1 in your submission contain map images which may be copyrighted. All PLOS content is published under the Creative Commons Attribution License (CC BY 4.0), which means that the manuscript, images, and Supporting Information files will be freely available online, and any third party is permitted to access, download, copy, distribute, and use these materials in any way, even commercially, with proper attribution. For these reasons, we cannot publish previously copyrighted maps or satellite images created using proprietary data, such as Google software (Google Maps, Street View, and Earth). For more information, see our copyright guidelines: http://journals.plos.org/plosone/s/licenses-and-copyright.

4.1.    You may seek permission from the original copyright holder of Figure 1 to publish the content specifically under the CC BY 4.0 license.

4.2.    If you are unable to obtain permission from the original copyright holder to publish these figures under the CC BY 4.0 license or if the copyright holder’s requirements are incompatible with the CC BY 4.0 license, please either i) remove the figure or ii) supply a replacement figure that complies with the CC BY 4.0 license. Please check copyright information on all replacement figures and update the figure caption with source information. If applicable, please specify in the figure caption text when a figure is similar but not identical to the original image and is therefore for illustrative purposes only.

Reviewers' comments:

Reviewer's Responses to Questions

**Comments to the Author**

1. Is the manuscript technically sound, and do the data support the conclusions?

Reviewer #1: No

2. Has the statistical analysis been performed appropriately and rigorously? 

Reviewer #1: Yes

3. Have the authors made all data underlying the findings in their manuscript fully available?

Reviewer #1: No

4. Is the manuscript presented in an intelligible fashion and written in standard English?

Reviewer #1: Yes

5. Review Comments to the Author

Reviewer #1: The manuscript entitled Effects of supplemental feeding on the gut microbiomes of Rocky Mountain elk in the Greater Yellowstone Ecosystem by Couch and colleagues attempts to describe the influence of supplemental feeding on the gastrointestinal microbiota of Rocky mountain Elk in the greater Yellowstone ecosystem, specifically at several feeding grounds in Wyoming. The paper is well written, samples appear to have been appropriately processed and analyses are mostly appropriate but are let down by several frail assumptions that negate the conclusions of the paper.

First and foremost is that samples are ground-collected fecal pellets. While I am keenly aware of the challenges of sampling wild animals, including Elk, it should be noted that microbiota detected in fecal pellets are significantly different from those detected in the rumen (e.g. Perea et al. DOI: 10.2527/jas.2016.1222), the paper I have provided as an example here may be important to reconciling this first issue as it not only shows these differences explicitly but also points to the importance of the distal gut (incl. fecal) microbiota to nutrition, measured therein as Feed efficiency. That being said, the discussion, including in the abstract on ruminal acidosis is not only speculative it is irrelevant to the study design.

Next, the collection of pellets from the ground will certainly carryover soil and other environmental microbes, thus conclusions around 'low-abundance bacterial genera' can not conclusively be attributed to differences in gut microbial populations. Finally, there is no knowledge of how many elk are represented by the pellets collected - they could theoretically represent a single animal at each sampling site, in which case the differences could simply reflect inter-individual variation.

Secondly, the sample sites are all within 8 - 12 miles of one another; given Elk can move this distance in a day, and that diet-induced transitions of the gut microbiota can take 3-4 weeks, how can the authors be certain the samples are reflective of elk strictly consuming the supplemental feeds at each feed station and not elk that have recently moved to one or other feed station?

Other concerns:

Sampling depth appears excellent, but coverage is not reported

The authors report differences in richness, but oddly do not look at alpha diversity, which is one of the most reliable measures of dysbiosis.

The F. necrophorum qPCR data is oddly tacked on, shows nothing of consequence and should be removed.

6. PLOS authors have the option to publish the peer review history of their article (what does this mean?). If published, this will include your full peer review and any attached files.

Reviewer #1: No

---

## [Author Response · Author response to Decision Letter 0]

20 Aug 2020

We are grateful to the editor and reviewers for their thorough and insightful comments and have addressed their concerns to the best of our abilities. Please note that, in addition to the changes suggested by the editor and reviewer, we have also implemented several minor changes suggested during an internal USGS review process. 

Editor:

Response: We have added an explanation to the methods section at lines 146-149 explaining why project-specific permits were not required. 

Response: We still intend to provide repository information prior to publication. 

4. We note that Figure1 in your submission contain map images which may be copyrighted. All PLOS content is published under the Creative Commons Attribution License (CC BY 4.0), which means that the manuscript, images, and Supporting Information files will be freely available online, and any third party is permitted to access, download, copy, distribute, and use these materials in any way, even commercially, with proper attribution. For these reasons, we cannot publish previously copyrighted maps or satellite images created using proprietary data, such as Google software (Google Maps, Street View, and Earth). For more information, see our copyright guidelines: http://journals.plos.org/plosone/s/licenses-and-copyright.

Response: The DEM and river/hydrology layers are USGS "owned" and the feedground locations are public information, so to the best of our knowledge, the whole figure is not copywrite-able and is in the public domain. The creator of the figure, Kimberly Szcodronski, is an author on the paper. We have added the text “map courtesy of the U.S. Geological Survey” to the end of the figure caption for clarity.

5. Please include captions for your Supporting Information files at the end of your manuscript, and update any in-text citations to match accordingly. 

Response: We have added the required captions and updated in-text citations to match. 

Reviewer #1: 

The manuscript entitled Effects of supplemental feeding on the gut microbiomes of Rocky Mountain elk in the Greater Yellowstone Ecosystem by Couch and colleagues attempts to describe the influence of supplemental feeding on the gastrointestinal microbiota of Rocky mountain Elk in the greater Yellowstone ecosystem, specifically at several feeding grounds in Wyoming. The paper is well written, samples appear to have been appropriately processed and analyses are mostly appropriate but are let down by several frail assumptions that negate the conclusions of the paper.

First and foremost is that samples are ground-collected fecal pellets. While I am keenly aware of the challenges of sampling wild animals, including Elk, it should be noted that microbiota detected in fecal pellets are significantly different from those detected in the rumen (e.g. Perea et al. DOI: 10.2527/jas.2016.1222), the paper I have provided as an example here may be important to reconciling this first issue as it not only shows these differences explicitly but also points to the importance of the distal gut (incl. fecal) microbiota to nutrition, measured therein as Feed efficiency. That being said, the discussion, including in the abstract on ruminal acidosis is not only speculative it is irrelevant to the study design.

Response: We understand the reviewer’s concerns regarding the biospatial delineations among GI tract microbiota, and we address this issue specifically regarding elk on line 347-350. However, the study we reference for ruminal acidosis-associated changes (line 339-340, Plaizier et al. 2017) reveals changes to the fecal microbiome in addition to the rumen microbiome, several of which are also observed in our study. Therefore, we regard the potential connection between the fecal microbiome and rumen acidosis in elk to be worthy of discussion and relevant to future research. 

Moreover, although Perea et al. identifies biospatial differences in the GI tract microbiota of lambs, the study also identifies feed efficiency-associated shifts that are consistent between the rumen and fecal microbiomes (e.g. OTU 3 was found to be more abundant in both the rumen and the feces of efficient vs inefficient lambs). Therefore, it is conceivable while sites along the GI tract host distinct microbial communities, individual taxa may exhibit trends that are consistent between GI tract regions.

Next, the collection of pellets from the ground will certainly carryover soil and other environmental microbes, thus conclusions around 'low-abundance bacterial genera' can not conclusively be attributed to differences in gut microbial populations. 

Response: Pellets were collected from snow-covered ground, which, while not sterile, likely precluded most soil microbes. We have added this information at line 130. Previous studies of snow surface microbiomes indicate extremely low biomass (<1,000 reads/ml snowmelt) and unique dominant taxa that were not present among the low-abundance taxa we discuss here (e.g. Michaud et al. 2014, DOI: 10.1371/journal.pone.0104505). 

Finally, there is no knowledge of how many elk are represented by the pellets collected - they could theoretically represent a single animal at each sampling site, in which case the differences could simply reflect inter-individual variation.

This is an excellent point, and we recognize that we did not give sufficient information to address this concern in the original manuscript. Although possible, it is extremely unlikely that duplicate samples were collected from the same individual at any given time point, because we collected very fresh samples that were still warm and not frozen, and outside temperatures generally far below freezing at time of collection. Given that the defecation rate of elk is approximately once per hour when elk are awake and active (Clinton Epps, personal communication), we believe we may safely assume that samples were not repeated within individuals at each time point. We have added a more thorough explanation of our methods and assumptions at lines 130-134.

Secondly, the sample sites are all within 8 - 12 miles of one another; given Elk can move this distance in a day, and that diet-induced transitions of the gut microbiota can take 3-4 weeks, how can the authors be certain the samples are reflective of elk strictly consuming the supplemental feeds at each feed station and not elk that have recently moved to one or other feed station?

Response: This is another valid concern that was not addressed sufficiently in the original manuscript. GPS collar data demonstrates that the vast majority of elk remain at a single location all winter, rarely moving more than 5 km away (see Appendix 2 of Merkle et al. 2017). We are therefore confident that samples reflect the diets being fed at each feedground. We have improved our explanation of movement at lines 113-115.

Other concerns:

Sampling depth appears excellent, but coverage is not reported

Based on the context, it is unclear whether the reviewer is referring to “coverage” in terms of redundancy per base (averaged across bacterial sequence variants), or total number of usable reads from the sequencing machine. We have added the latter value to the manuscript at line 173, as it is likely to be of more general interest and can be used to estimate the former value if desired.

The authors report differences in richness, but oddly do not look at alpha diversity, which is one of the most reliable measures of dysbiosis.

Response: Species richness is itself a measure of alpha diversity, and arguably more transparent and easier to interpret than indices that incorporate evenness. Preliminary analyses demonstrated that Shannon diversity exhibited very similar patterns to observed richness. However, we opted to include only richness as we were primarily interested in beta diversity and composition, and including additional metrics of alpha diversity did not assist in addressing our hypotheses.

The F. necrophorum qPCR data is oddly tacked on, shows nothing of consequence and should be removed.

Response: We agree that the qPCR results appear somewhat tangential to the overall findings of the paper, however, the original intent of the study was to identify correlations between F. necrophorum abundance and microbiome community variation, and it was with this intent that we developed and validated this qPCR assay. The protocol we developed is unlikely to be published if not included in this paper, and we believe its publication will be valuable to other researchers studying F. necrophorum in elk. Given that PLOS One evaluates research on the basis of scientific validity rather than impact, we hope that we will be permitted to include the F. necrophorum protocol and results in this publication.

---

## [Decision Letter · Decision Letter 1]

15 Oct 2020

PONE-D-20-13745R1

Effects of supplemental feeding on the gut microbiomes of Rocky Mountain elk in the Greater Yellowstone Ecosystem

PLOS ONE

Dear Dr. Couch,

Thank you for submitting your manuscript to PLOS ONE. After careful consideration, we have decided that your manuscript does not meet our criteria for publication and must therefore be rejected.

Specifically:

MAJOR ISSUES RAISED IN THE ORIGINAL REVIEW REMAIN AS IT SEEMS THAT AUTHORS DID NOT SERIOUSLY CONSIDER THEM.

I am sorry that we cannot be more positive on this occasion, but hope that you appreciate the reasons for this decision.

Yours sincerely,

Juan J Loor

Academic Editor

PLOS ONE

Reviewers' comments:

Reviewer's Responses to Questions

**Comments to the Author**

1. If the authors have adequately addressed your comments raised in a previous round of review and you feel that this manuscript is now acceptable for publication, you may indicate that here to bypass the “Comments to the Author” section, enter your conflict of interest statement in the “Confidential to Editor” section, and submit your "Accept" recommendation.

Reviewer #1: (No Response)

2. Is the manuscript technically sound, and do the data support the conclusions?

Reviewer #1: No

3. Has the statistical analysis been performed appropriately and rigorously? 

Reviewer #1: No

4. Have the authors made all data underlying the findings in their manuscript fully available?

Reviewer #1: No

5. Is the manuscript presented in an intelligible fashion and written in standard English?

Reviewer #1: Yes

6. Review Comments to the Author

Reviewer #1: Rather than addressing several significant concerns, the authors have elected to be dismissive leaving a manuscript that lacks rigor with conclusions built on multiple unprovable assumptions given the data provided and study design. Collectively it is hard to see what value it adds to the literature. This is perhaps best exemplified by the persistent inclusion of F. necrophorum qPCR data that, by the authors own admission are irrelevant because it is unlikely to be published elsewhere. For future it should also be noted that diversity by definition is a measure of both richness and evenness, therefore richness is not a complete measure of alpha-diversity, further alpha-diversity, including evenness is a more faithful indicator of dysbiosis which was the point the author is trying to speak to.

7. PLOS authors have the option to publish the peer review history of their article (what does this mean?). If published, this will include your full peer review and any attached files.

Reviewer #1: No

- - - - -

---

## [Author Response · Author response to Decision Letter 1]

15 Nov 2020

Response to Academic Editor and Reviewer:

Please see below for point-by-point responses to the editor and reviewer comments accompanying the rejection of our manuscript. Editor and reviewer comments are copied and pasted exactly as they were written, with our responses provided below. 

The below text is copied directly (including capitalization and punctuation) from the rejection email we received from Dr. Juan J. Loor, the Academic Editor who handled this manuscript:

"Dear Dr. Couch,

Thank you for submitting your manuscript to PLOS ONE. After careful consideration, we have decided that your manuscript does not meet our criteria for publication and must therefore be rejected.

Specifically:

MAJOR ISSUES RAISED IN THE ORIGINAL REVIEW REMAIN AS IT SEEMS THAT AUTHORS DID NOT SERIOUSLY CONSIDER THEM.

I am sorry that we cannot be more positive on this occasion, but hope that you appreciate the reasons for this decision.

Yours sincerely,

Juan J Loor

Academic Editor

PLOS ONE"

Our response to this comment:

While we appreciate the time and effort the Academic Editor devoted to evaluating this manuscript, their specific reasons for rejection are unclear based on the limited justification provided. We are concerned that, because both the first and second rounds of reviews were conducted by a single reviewer (apparently the same individual in both instances), the editor’s decision was based on insufficient information. Additionally, we do not agree with the editor’s statement that we did not take reviewer concerns seriously. While we respectfully disagreed with a small number of the reviewer’s initial comments, we considered their arguments very seriously and attempted to thoroughly address and explain the logic underlying our disagreements (please refer to our initial reviewer response). In the second round of review, the primary concern of the reviewer seems to be our difference of opinion regarding the inclusion of methodology and null results from an assay we developed and validated as part of this study. To our understanding, a primary goal of PLOS ONE is to evaluate research based on scientific merit rather than impact or significance of the results. We therefore request that the decision to reject this manuscript, which appears to conflict with the mission of PLOS ONE, be reconsidered.

The reviewer’s only comment from this round of review is shown below:

"Rather than addressing several significant concerns, the authors have elected to be dismissive leaving a manuscript that lacks rigor with conclusions built on multiple unprovable assumptions given the data provided and study design. Collectively it is hard to see what value it adds to the literature. This is perhaps best exemplified by the persistent inclusion of F. necrophorum qPCR data that, by the authors own admission are irrelevant because it is unlikely to be published elsewhere. For future it should also be noted that diversity by definition is a measure of both richness and evenness, therefore richness is not a complete measure of alpha-diversity, further alpha-diversity, including evenness is a more faithful indicator of dysbiosis which was the point the author is trying to speak to."

Our response to this comment:

The reviewer makes three main points: (1) we were overly dismissive of the reviewer’s concerns, (2) our manuscript lacks rigor with conclusions based on unprovable assumptions, (3) we included methodology and null findings from a qPCR assay developed as part of this study, and (4) we opted to use observed richness as a measure of alpha diversity. We will address each point below:

1. Perhaps the validity of this concern can be best evaluated by referring our original response to the reviewer (attached). While we respectfully disagreed with the reviewer on several points, we took their comments very seriously and attempted to provide full, transparent, and logical justification for these disagreements. We request that the individual/s reviewing this resubmission refer to our original reviewer response to evaluate the validity of this concern. 

2. The reviewer does not provide specific instances of unprovable assumptions or lack of rigor. The specific concerns raised by the reviewer (points 3 & 4) do not relate to rigor or assumptions, but rather differences of opinion regarding the value of including null results and the relative merit of different measures of alpha diversity. However, as our original responses to points 3&4 were unsatisfactory to the reviewer, we have attempted to revise our manuscript to better address their concerns (see below for specifics). 

3. As explained in our original rebuttal letter, we wish to retain methodology and null findings from our F. necrophorum qPCR assay because we believe its publication will be valuable to other researchers studying this pathogen. Given that PLOS ONE evaluates research on the basis of scientific validity rather than perceived impact, we hope to include the methodology and findings in this manuscript. 

We have attempted to better integrate the F. necrophorum methods, results, and significance more cohesively into the manuscript by revising/adding explanation of methods at lines 166-168 and results at lines 227-230. The importance of our findings are discussed at lines 357-365. However, if a second reviewer also recommends that we remove the qPCR material, we would be happy to reconsider. 

4. The reviewer objects strongly to our use of observed richness as a measure of alpha diversity rather than an index that incorporates evenness. According to the reviewer, alpha diversity by definition includes a measure of evenness. However, Whittaker’s original definition of alpha diversity specifically states that it is a measure of the number of species in a locality or habitat (Whittaker 1960, Ecological Monographs), not to be confused with Fisher’s alpha, which is a diversity index. In microbiome research, observed richness normalized by sampling depth is a commonly used measure of alpha diversity. Evenness estimates from amplicon sequence data can be skewed by sequencing bias and data aggregation, and diversity indices that include evenness are more difficult to interpret than observed richness. Additionally, the reviewer states that evenness is one of the most reliable indicators of dysbiosis, but they fail to present any references to support this claim. To our understanding, this argument does not align with the literature. In many studies of the gut microbiome, reduction in species richness is a primary indicator of dysbiosis, though the reviewer is correct that this pattern frequently coincides with reduction in evenness and other alpha diversity indices. We therefore respectfully disagreed with the reviewer’s recommendation during the first round of reviews. 

However, as our justification for omitting alpha diversity metrics was unsatisfactory to the reviewer, we have now added Inverse Simpson and Shannon diversity comparisons to the cross-sectional analysis at lines 182-193 and in Figure 2. Results from this analysis are consistent with the previously reported observed richness results (lines 223-227, Figure 2). Therefore, inclusion of the alpha diversity indices did not alter our conclusions.

---

## [Decision Letter · Decision Letter 2]

18 Jan 2021

PONE-D-20-13745R2

Effects of supplemental feeding on the gut microbiomes of Rocky Mountain elk in the Greater Yellowstone Ecosystem

PLOS ONE

Dear Dr. Couch,

Thank you for submitting your manuscript to PLOS ONE. After careful consideration, we feel that it has merit but does not fully meet PLOS ONE’s publication criteria as it currently stands. Therefore, we invite you to submit a revised version of the manuscript that addresses the points raised during the review process.

Thank you for your patience during this unusually long and complicated review process.  I received comments from the previous round of review from a single reviewer, and regardless of the actual review itself, I do not feel that a single reviewer is sufficient to assess any manuscript.  Thus, I sought additional reviewers, and because of the dispute of the previous editorial decision, I solicited three additional reviewers to complement the original review.  I also carefully considered the previous response to reviewer comments, and the justification provided by the authors.

Collectively, the fours reviews have completely different recommendations, and so I carefully compiled the comments in each to determine the amount and seriousness of the recommendations. Overall, there was interest in the information presented here, and an understanding that microbiome work with wild ruminants is often logistically difficult.  After weighing these review comments, I decided on "major revision" due to the large number of comments made and because some of them require some consideration, but not because the comments indicate that a complete restructuring is needed.  Most of the new comments seem straightforward, so I am commenting only on the two major points of contention.

With regards to diversity metrics, I tend to agree with the reviewers and feel that the way that most metrics are taught, presented, and interpreted make them seem more interchangable than they are. The authors likely know all this, and I include the explanation here not to pander but to explain my view, which I believe is shared by multiple reviewers.  To the point about evenness, this information is often incorporated into calculations such as Shannon, or Simpsons, in some way, but I find that it is possible to obscure trends in either richness or evenness by combining them into a holistic diversity metric.  I used to use Shannon's all the time, and now I find it is more informative to use richness and evenness specifically, because they reveal more important trends.  For example, whether all bacteria are equally affected by a treatment which reduces richness, or only certain members. Thus, the authors have previously tried to address the concern over diversity metrics by adding additional ones, but I think this point can best be settled by the authors verifying that the diversity metrics they have selected indeed provide the information they find most pertinent to their study.

With regards to the qPCR data for testing one particular bacteria, the reviewers had mixed opinions on its usefulness, but I think this can best be addressed by adding a few extra sentences of justification for why this particular bacteria was selected, when there are many potential pathogens the authors could have chosen.  It seems like the authors chose this species because it is of concern in feedlot sheep, and may also be of concern in feedlot-raised elk, as well.  If this is the case, the authors should more explicitly state this reasoning.

We look forward to receiving your revised manuscript.

Kind regards,

Suzanne L. Ishaq, PhD

Academic Editor

PLOS ONE

and

Ibukun Ogunade

Academic Editor

PLOS ONE

Journal Requirements:

2.  Please upload a copy of Supporting Information Supplementary figure S1: which you refer to in your text on line 571-572.

Reviewers' comments:

Reviewer's Responses to Questions

**Comments to the Author**

1. If the authors have adequately addressed your comments raised in a previous round of review and you feel that this manuscript is now acceptable for publication, you may indicate that here to bypass the “Comments to the Author” section, enter your conflict of interest statement in the “Confidential to Editor” section, and submit your "Accept" recommendation.

Reviewer #2: (No Response)

Reviewer #3: (No Response)

Reviewer #4: All comments have been addressed

2. Is the manuscript technically sound, and do the data support the conclusions?

Reviewer #2: Yes

Reviewer #3: Partly

Reviewer #4: Yes

3. Has the statistical analysis been performed appropriately and rigorously? 

Reviewer #2: Yes

Reviewer #3: I Don't Know

Reviewer #4: Yes

4. Have the authors made all data underlying the findings in their manuscript fully available?

Reviewer #2: Yes

Reviewer #3: Yes

Reviewer #4: Yes

5. Is the manuscript presented in an intelligible fashion and written in standard English?

Reviewer #2: Yes

Reviewer #3: Yes

Reviewer #4: Yes

6. Review Comments to the Author

Reviewer #2: This study characterized the changes in the gut microbiomes of wild elk under different supplemental feeding regimes and provides useful information on how alfalfa pellets supplemental feeding can alter the native gut microbiome of wild animals and impact their health. The authors carried out comprehensive analyses of the microbiome data.

For the two major disputes from the previous review. The following are my comments/recommendations.

1. In my opinion, the authors have adequately addressed the alpha-diversity query. There is some terminology confusion between “diversity” and “alpha-diversity”, particularly in how they are used in macro and microbial ecology, but I don’t think it’s fair to overly penalize the authors for such a minor terminology difference that is widespread in the field.

I do think it’s always a good idea to look at multiple alpha-diversity metrics and it’s more convincing now that they have added in two other metrics (which both represent evenness to some degree).

The degree to which richness or evenness better capture dysbiosis is definitely a contentious point. I think the previous reviewer may actually be right regarding the importance of evenness, but that’s just my opinion and far from accepted in the field - richness is the more commonly compared metric.

In conclusion, I think what the authors provided have sufficiently address the issue.

2. For the qPCR measurement of the F. necrophorum, I think it will be okay to keep. However, the writing of this experiment in the Result is not clear. I am copying exactly what is the manuscript below starting from Line 227. “The F. necrophorum qPCR assay did not detect either strain in any of the elk fecal samples, save for a single sample that amplified a low amount of F. necrophorum funduliforme, suggesting that these species are rarely shed in elk feces”

First, the period at the end of the sentence is missing. More importantly, I don’t think this sentence is correct or clear. This sentence needs to be revised for clarify.

Other than the two points above, I am in favor of accepting this manuscript for publication as it certainly contains helpful information for the field of wild life microbiome studies.

Reviewer #3: Manuscript Number: PONE-D-20-13745R2

Manuscript Title: Effects of supplemental feeding on the gut microbiomes of Rocky Mountain elk in the Greater Yellowstone Ecosystem

The objective of this current study was to assess the impact of supplemental winter feeding on the gut microbiome of Rocky Mountain elk in western Wyoming exploring the potential implications of feeding on the elk population health and disease. This is a descriptive study of the bacterial diversity in fecal samples from a large number of wild, free-ranging elk on either natural pasture or fed supplemental feed at 14 different locations and timepoints (8-23 samples per time point, 273 samples in total according to Table 1).

Title

The title should reflect the fact that fecal samples were analyzed – not “gut” samples – as noninvasive sampling was employed without sacrificing animals and sampling the different sections of their digestive tract. This wording should be corrected throughout the manuscript to avoid misunderstandings and to be clear about the identity of the samples. The title should also communicate that only the bacterial microbiome of the feces was analyzed. The ruminant rumen and hindgut microbiome includes both bacteria, ciliates, anaerobic fungi and methanogens.

Elk are intermediate ruminants

As far as I can see, the word “ruminant” is only mentioned twice in the manuscript, fist at the end of the introduction when describing the work to develop an elk-specific assay to assess the abundance of Fusobacterium necrophorum, and then one more time at the end of the discussion when talking about this widespread commensal in elk feces. The fact that the elk is a ruminant is a key feature to understand it´s gut microbiome, and the effect of diet and supplemental feeding on the gut microbiome and the health of the host animal. I miss a discussion on the physiological and anatomical adaptations of these ruminants in the manuscript. Please include a description of their digestive tract with reference to the pioneering work by Hofmann, and the symbiotic microbial digestion of their herbivorous diet in the reticulorumen and the hindgut. Ruminants on natural pasture with seasonal changes in appetite / food intake and exposed to seasonal changes in diet composition and chemistry show seasonal changes in their symbiotic rumen and hindgut microflora (e.g. the high-arctic Svalbard reindeer Rangifer tarandus platyrhynchus (Orpin et al 1985 https://pubmed.ncbi.nlm.nih.gov/4026289/ )). As underlined in the abstract of this manuscript the interplay between diet, the microbiome, and the host health highlights the need to understand the consequences of supplemental feeding on the microbiomes of free-ranging “populations”. This has been studied in e.g. the semi-domesticated intermediate ruminant reindeer (Rangifer tarandus tarandus) (see review by Sundset et al. 2015 Encyclopedia of Metagenomics https://link.springer.com/referenceworkentry/10.1007%2F978-1-4899-7475-4_664 ). Supplemental feeding in periods when access to pasture is poor and the animals may have starved is challenging and may result in rumen malfunction e.g. rumen acidosis.

Materials, methods, data

• Age and sex of the animals sampled are not presented / known (both are known to influence the composition of the gut microbiome) but may be discussed in relation to these data.

• PCR and sequencing of the 16S V4 region was performed. Please include the primer sets used under the method section on sample sequencing.

• According to Table 1, 273 samples were collected? It is stated in the method section that the samples were split equally between two MiSeq runds including 315 elk fecal samples and including the 199 samples used in this study. This is not clear to me. What happened to the other samples? What were they used for? And then below you state that a total of 22,620,453 reads were obtained from the 315 samples? Did you use 315 or 199 samples? Or 273 samples?

• Several studies exist on the microbiome in the gastrointestinal tract of wild ruminants. It would be nice to see a comparison between this current dataset from the Rocky Mountain elk feces and e.g. colonic samples from North American moose – the largest browsing ruminant of the deer family (Ishaq & Wright, 2012, BMC Microbiology 12) or datasets generated from the feces of other ruminants (wild or domestic).

• Also, as only feces and not samples from the digestive tract were obtained, please discuss this aspect with reference to other papers comparing the feces microbiome to that of e.g. the colonic microbiome in ruminants. What are the pros and cons for using fecal samples instead of e.g. rumen samples when investigating the potential implications of feeding on the elk population health and disease?

• Did you see any health or disease problems among the animals sampled in this current study?

Fusobacterium necrophorum

F. necrophorum is an opportunistic pathogen found in the digestive tract of both humans and animals, known to cause necrotic conditions including liver abscesses and foot rot in ruminants, with the subspecies F. n. necrophorum (biotype A) most virulent and isolated more frequently from infections (Nagaraja et al. 2005, Veterinary anaerobes and diseases 11: 239-246). The current study presents a qPCR essay for F. necrophorum and also screened the large number of fecal samples collected from elk on both natural pasture and eating supplemental feed at different locations and different times. The qPCR essay did not detect either of the two strains in any of the samples except one with low amounts of F.n. funduliforme (biotype B). Hence, this study indicates that these populations of wild elk in the Rocky Mountains rarely shed F. necrophorum. Although, F. necrophorum has indeed previously been isolated from elk (4 isolates from footrot) by Clifton et al. (2018) Veterinary Microbiology 213: 108-113.

I agree with the authors that the findings from this current study are valuable to other researchers studying this pathogen and support their publication along with the microbiome data.

Does rumen acidosis effect the fecal microbiome? And is it likely that these animals were challenged with rumen acidosis?

Discussing their findings that Proteobacteria and Verrucomicrobia were reduced in elk fed alfalfa pellets the authors refer to the study by Plaizier et al. (2016) on the effect of a grain-based subacute ruminal acidosis challenge on the rumen digesta and feces microbiota (ref 60). Plaizier et al. concluded that also the bacterial community composition in the feces was affected by the rumen acidosis challenge (altering the lower taxonomical level), but they did not identify any bacterial taxa in the feces that could be used for accurate and non-invasive diagnosis of rumen acidosis. High intakes of rapidly digestible carbohydrates such as barley or other cereals are the primary cause of rumen acidosis in ruminants. In acute situations this may result in death. I could not access ref 61 (Hattel et al. 2007) showing that rumen acidosis is the leading cause of death among captive elk (Cervus Elaphus) in Pennsylvania – but the captive elks in this previous study may have received a different diet with a higher content of soluble carbohydrates compared to the wild Rocky Mountain elk? The chemical / nutrient analysis of the supplemental feeds given to the Rocky Mountain elk in this current study however showed that the pelleted alfalfa was low in soluble carbohydrates (and high in proteins) and would perhaps not be expected to cause rumen acidosis?

Reviewer #4: (No Response)

7. PLOS authors have the option to publish the peer review history of their article (what does this mean?). If published, this will include your full peer review and any attached files.

Reviewer #2: No

Reviewer #3: **Yes: **Monica A. Sundset

Reviewer #4: No

---

## [Author Response · Author response to Decision Letter 2]

18 Mar 2021

Editor comments:

Comment: With regards to diversity metrics, I tend to agree with the reviewers and feel that the way that most metrics are taught, presented, and interpreted make them seem more interchangable than they are. The authors likely know all this, and I include the explanation here not to pander but to explain my view, which I believe is shared by multiple reviewers. To the point about evenness, this information is often incorporated into calculations such as Shannon, or Simpsons, in some way, but I find that it is possible to obscure trends in either richness or evenness by combining them into a holistic diversity metric. I used to use Shannon's all the time, and now I find it is more informative to use richness and evenness specifically, because they reveal more important trends. For example, whether all bacteria are equally affected by a treatment which reduces richness, or only certain members. Thus, the authors have previously tried to address the concern over diversity metrics by adding additional ones, but I think this point can best be settled by the authors verifying that the diversity metrics they have selected indeed provide the information they find most pertinent to their study.

Response: We have attempted to clarify why the diversity metrics we have selected provide the most pertinent information to our study by adding the following text at lines 183-184 explaining our choice of indices: “Inverse Simpson and Shannon diversity indices, both of which incorporate taxonomic evenness in addition to richness, were also calculated in phyloseq [47] to assess whether alpha diversity results were especially sensitive to changes in rare taxa (Shannon) or common taxa (Inverse Simpson).”

Comment: With regards to the qPCR data for testing one particular bacteria, the reviewers had mixed opinions on its usefulness, but I think this can best be addressed by adding a few extra sentences of justification for why this particular bacteria was selected, when there are many potential pathogens the authors could have chosen. It seems like the authors chose this species because it is of concern in feedlot sheep, and may also be of concern in feedlot-raised elk, as well. If this is the case, the authors should more explicitly state this reasoning.

Response: The rationale for selecting Fusobacterium necrophorum was because it is a pathogen of concern for Wyoming elk. We have clarified this at line 118.

Reviewer #2: This study characterized the changes in the gut microbiomes of wild elk under different supplemental feeding regimes and provides useful information on how alfalfa pellets supplemental feeding can alter the native gut microbiome of wild animals and impact their health. The authors carried out comprehensive analyses of the microbiome data.

For the two major disputes from the previous review. The following are my comments/recommendations.

1. In my opinion, the authors have adequately addressed the alpha-diversity query. There is some terminology confusion between “diversity” and “alpha-diversity”, particularly in how they are used in macro and microbial ecology, but I don’t think it’s fair to overly penalize the authors for such a minor terminology difference that is widespread in the field.

I do think it’s always a good idea to look at multiple alpha-diversity metrics and it’s more convincing now that they have added in two other metrics (which both represent evenness to some degree).

The degree to which richness or evenness better capture dysbiosis is definitely a contentious point. I think the previous reviewer may actually be right regarding the importance of evenness, but that’s just my opinion and far from accepted in the field - richness is the more commonly compared metric.

In conclusion, I think what the authors provided have sufficiently address the issue.

2. For the qPCR measurement of the F. necrophorum, I think it will be okay to keep. However, the writing of this experiment in the Result is not clear. I am copying exactly what is the manuscript below starting from Line 227. “The F. necrophorum qPCR assay did not detect either strain in any of the elk fecal samples, save for a single sample that amplified a low amount of F. necrophorum funduliforme, suggesting that these species are rarely shed in elk feces”

First, the period at the end of the sentence is missing. More importantly, I don’t think this sentence is correct or clear. This sentence needs to be revised for clarify.

Response: We have made the following revision for clarity at lines 231-233: “F. necrophorum funduliforme was identified in only a single fecal sample, and F. necrophorum necrophorum was not detected in any samples, suggesting that these species are rarely or never shed in elk feces.”

Other than the two points above, I am in favor of accepting this manuscript for publication as it certainly contains helpful information for the field of wild life microbiome studies.

Reviewer #3: Manuscript Number: PONE-D-20-13745R2

Manuscript Title: Effects of supplemental feeding on the gut microbiomes of Rocky Mountain elk in the Greater Yellowstone Ecosystem

The objective of this current study was to assess the impact of supplemental winter feeding on the gut microbiome of Rocky Mountain elk in western Wyoming exploring the potential implications of feeding on the elk population health and disease. This is a descriptive study of the bacterial diversity in fecal samples from a large number of wild, free-ranging elk on either natural pasture or fed supplemental feed at 14 different locations and timepoints (8-23 samples per time point, 273 samples in total according to Table 1).

Comment: Title

The title should reflect the fact that fecal samples were analyzed – not “gut” samples – as noninvasive sampling was employed without sacrificing animals and sampling the different sections of their digestive tract. This wording should be corrected throughout the manuscript to avoid misunderstandings and to be clear about the identity of the samples. The title should also communicate that only the bacterial microbiome of the feces was analyzed. The ruminant rumen and hindgut microbiome includes both bacteria, ciliates, anaerobic fungi and methanogens.

Response: The title has been changed to “Effects of supplemental feeding on the fecal bacterial communities of Rocky Mountain elk in the Greater Yellowstone Ecosystem”. 

Comment: Elk are intermediate ruminants

As far as I can see, the word “ruminant” is only mentioned twice in the manuscript, fist at the end of the introduction when describing the work to develop an elk-specific assay to assess the abundance of Fusobacterium necrophorum, and then one more time at the end of the discussion when talking about this widespread commensal in elk feces. The fact that the elk is a ruminant is a key feature to understand it´s gut microbiome, and the effect of diet and supplemental feeding on the gut microbiome and the health of the host animal. I miss a discussion on the physiological and anatomical adaptations of these ruminants in the manuscript. Please include a description of their digestive tract with reference to the pioneering work by Hofmann, and the symbiotic microbial digestion of their herbivorous diet in the reticulorumen and the hindgut. 

Response: A brief description of the elk digestive system has been included at lines 347-352. 

Comment: Ruminants on natural pasture with seasonal changes in appetite / food intake and exposed to seasonal changes in diet composition and chemistry show seasonal changes in their symbiotic rumen and hindgut microflora (e.g. the high-arctic Svalbard reindeer Rangifer tarandus platyrhynchus (Orpin et al 1985 https://pubmed.ncbi.nlm.nih.gov/4026289/ )). As underlined in the abstract of this manuscript the interplay between diet, the microbiome, and the host health highlights the need to understand the consequences of supplemental feeding on the microbiomes of free-ranging “populations”. This has been studied in e.g. the semi-domesticated intermediate ruminant reindeer (Rangifer tarandus tarandus) (see review by Sundset et al. 2015 Encyclopedia of Metagenomics https://link.springer.com/referenceworkentry/10.1007%2F978-1-4899-7475-4_664 ). Supplemental feeding in periods when access to pasture is poor and the animals may have starved is challenging and may result in rumen malfunction e.g. rumen acidosis.

Materials, methods, data

• Comment: Age and sex of the animals sampled are not presented / known (both are known to influence the composition of the gut microbiome) but may be discussed in relation to these data.

Response: We have added an acknowledgement of this to the discussion at lines 365-366. 

Comment: PCR and sequencing of the 16S V4 region was performed. Please include the primer sets used under the method section on sample sequencing.

Response: Primers have been included at line 164

Comment: According to Table 1, 273 samples were collected? It is stated in the method section that the samples were split equally between two MiSeq runds including 315 elk fecal samples and including the 199 samples used in this study. This is not clear to me. What happened to the other samples? What were they used for? And then below you state that a total of 22,620,453 reads were obtained from the 315 samples? Did you use 315 or 199 samples? Or 273 samples?

Response: The original number of 199 in the methods section was incorrect. The total number of samples used in this study was 282. This has been corrected in the text. The additional samples from the sequencing run were from a separate study on elk disease or did not meet the minimum sequencing depth requirement.

Comment: Several studies exist on the microbiome in the gastrointestinal tract of wild ruminants. It would be nice to see a comparison between this current dataset from the Rocky Mountain elk feces and e.g. colonic samples from North American moose – the largest browsing ruminant of the deer family (Ishaq & Wright, 2012, BMC Microbiology 12) or datasets generated from the feces of other ruminants (wild or domestic).

Response: Although we agree with the reviewer that such a comparison would be useful, it is somewhat out of the scope of our hypotheses. However, sequencing data will be made publicly available for future comparisons. 

Comment: Also, as only feces and not samples from the digestive tract were obtained, please discuss this aspect with reference to other papers comparing the feces microbiome to that of e.g. the colonic microbiome in ruminants. What are the pros and cons for using fecal samples instead of e.g. rumen samples when investigating the potential implications of feeding on the elk population health and disease?

Response: We have included the following text at lines 371-375: “In domestic ruminants, the foregut microbiome has higher richness and may be more responsive to feed changes than the fecal microbiome (Lourenco et al. 2020). Therefore, while the fecal microbiomes of ruminants are easily sampled noninvasively, they represent only a subset of the complex and variable gastrointestinal tract microbiome and must be interpreted with care.”

Comment: Did you see any health or disease problems among the animals sampled in this current study?

Response: No – we did not observe any obvious signs of disease. However, because disease data was not included in this study, we cannot confidently assert that disease was absent.

Comment: Fusobacterium necrophorum

F. necrophorum is an opportunistic pathogen found in the digestive tract of both humans and animals, known to cause necrotic conditions including liver abscesses and foot rot in ruminants, with the subspecies F. n. necrophorum (biotype A) most virulent and isolated more frequently from infections (Nagaraja et al. 2005, Veterinary anaerobes and diseases 11: 239-246). The current study presents a qPCR essay for F. necrophorum and also screened the large number of fecal samples collected from elk on both natural pasture and eating supplemental feed at different locations and different times. The qPCR essay did not detect either of the two strains in any of the samples except one with low amounts of F.n. funduliforme (biotype B). Hence, this study indicates that these populations of wild elk in the Rocky Mountains rarely shed F. necrophorum. Although, F. necrophorum has indeed previously been isolated from elk (4 isolates from footrot) by Clifton et al. (2018) Veterinary Microbiology 213: 108-113.

I agree with the authors that the findings from this current study are valuable to other researchers studying this pathogen and support their publication along with the microbiome data.

Does rumen acidosis effect the fecal microbiome? And is it likely that these animals were challenged with rumen acidosis?

Discussing their findings that Proteobacteria and Verrucomicrobia were reduced in elk fed alfalfa pellets the authors refer to the study by Plaizier et al. (2016) on the effect of a grain-based subacute ruminal acidosis challenge on the rumen digesta and feces microbiota (ref 60). Plaizier et al. concluded that also the bacterial community composition in the feces was affected by the rumen acidosis challenge (altering the lower taxonomical level), but they did not identify any bacterial taxa in the feces that could be used for accurate and non-invasive diagnosis of rumen acidosis. 

Response: The reviewer’s comment reflects the content of the abstract of Plaizier et al (ref 60), however, the main text contains the following additional information: “In the feces, the SARA challenge did not affect the relative abundances of Firmicutes and Bacteroidetes, but it decreased those of Tenericutes, Proteobacteria, Cyanobacteria, Verrucomicrobia, and Fibrobacteres, and decreased those of Spirochaetes and Actinobacteria.” Therefore, though these taxa may not be sufficient for accurate diagnosis of rumen acidosis, we suggest that the relationships described in the current study and by Plaizier et al. may be worth exploring further as potential fecal indicators of rumen acidosis. Additionally, the Plaizier et al. study identifies a reduction in Firmicutes in the rumen but not in the feces of cattle with rumen acidosis. In our study, Firmicutes increased in the feces of alfalfa-fed elk, but we have no information regarding the rumen microbiome. However, due to the differences in host physiology and study design between our current study and Plazier et al., we believe this finding warrants further exploration. 

Comment: High intakes of rapidly digestible carbohydrates such as barley or other cereals are the primary cause of rumen acidosis in ruminants. In acute situations this may result in death. I could not access ref 61 (Hattel et al. 2007) showing that rumen acidosis is the leading cause of death among captive elk (Cervus Elaphus) in Pennsylvania – but the captive elks in this previous study may have received a different diet with a higher content of soluble carbohydrates compared to the wild Rocky Mountain elk? The chemical / nutrient analysis of the supplemental feeds given to the Rocky Mountain elk in this current study however showed that the pelleted alfalfa was low in soluble carbohydrates (and high in proteins) and would perhaps not be expected to cause rumen acidosis?

Response: The reviewer is correct that the pelleted alfalfa does not contain excessive amounts of soluble carbohydrates and it is high in protein. However, it is also low in fiber relative to the other feed types. Previous research has demonstrated that alfalfa pellets reduce rumen pH relative to alfalfa hay in cattle (Khafipoor et al. 2007 J Animal Sci). We agree with the reviewer that we cannot conclusively diagnose rumen acidosis, and that further research is needed to understand whether feeding concentrated alfalfa pellets induces rumen acidosis in elk. However, we also believe that findings from our study should motivate future research to clarify potential links between concentrated feed and rumen biochemistry in elk, as these questions could have important conservation and management implications. These points are summarized in the text at lines 357-368.

Regarding ref 61, Hattel et al. determined the cause of death for 5 out of a herd of 65 elk was rumen acidosis, and therefore rumen acidosis was among the top 5 causes of mortality.

---

## [Editor Report · Decision Letter 3]

22 Mar 2021

Effects of supplemental feeding on the fecal bacterial communities of Rocky Mountain elk in the Greater Yellowstone Ecosystem

PONE-D-20-13745R3

Dear Dr. Couch,

We’re pleased to inform you that your manuscript has been judged scientifically suitable for publication and will be formally accepted for publication once it meets all outstanding technical requirements. We greatly appreciate the time and effort that the authors have put into this manuscript during this difficult review process, and especially your willingness to help reviewers best understand your results and interpretation. 

Kind regards,

Suzanne L. Ishaq, PhD

Academic Editor

PLOS ONE
---

## [Editor Report · Acceptance letter]

25 Mar 2021

PONE-D-20-13745R3 

Effects of supplemental feeding on the fecal bacterial communities of Rocky Mountain elk in the Greater Yellowstone Ecosystem 

Dear Dr. Couch:

I'm pleased to inform you that your manuscript has been deemed suitable for publication in PLOS ONE. Congratulations! Your manuscript is now with our production department. 

Kind regards, 

on behalf of

Dr. Suzanne L. Ishaq 

Academic Editor

PLOS ONE